# Learning from Videos for 3D World: Enhancing MLLMs with 3D Vision Geometry Priors

**Duo Zheng**[*], **Shijia Huang**[*], **Yanyang Li, Liwei Wang**[♯]

The Chinese University of Hong Kong

https://lavi-lab.github.io/VG-LLM

## Abstract

Previous research has investigated the application of Multimodal Large Language Models (MLLMs) in understanding 3D scenes by interpreting them as videos. These approaches generally depend on comprehensive 3D data inputs, such as point clouds or reconstructed Bird's-Eye View (BEV) maps. In our research, we advance this field by enhancing the capability of MLLMs to understand and reason in 3D spaces directly from video data, without the need for additional 3D input. We propose a novel and efficient method called the Video-3D Geometry Large Language Model (VG LLM). Our approach utilizes a 3D visual geometry encoder to extract 3D prior information from video sequences. This information is then integrated with visual tokens and input into the MLLM. Extensive experiments have shown that our method has achieved substantial improvements in various tasks related to 3D scene understanding and spatial reasoning, all directly learned from video sources. Impressively, our 4B model, which does not rely on explicit 3D data inputs, achieves competitive results compared to existing state-of-the-art methods, and even surpasses the Gemini-1.5-Pro in the VSI-Bench evaluations.

## 1 Introduction

Rapid advancements and impressive performance of Multimodal Large Language Models (MLLMs) [2, 24, 26, 29, 32, 39, 40, 43] have driven their applications in various fields, such as 3D scene understanding [11, 20, 21, 22, 35, 58, 63], vision-language-action models [3, 25, 36, 47], embodied navigation [55, 59, 61], and learning 3D knowledge from videos [35, 55, 58].

Efforts [35, 41, 58] have been made to improve the 3D spatial understanding capability of MLLMs by considering scenes as video sequences. For example, Video-3D LLM [41, 58] injects 3D coordinates into visual features at patch level to improve 3D perception. GPT4Scene [35] leverages BEV maps [28] rendered from reconstructed 3D point clouds for global awareness. However, a shared limitation of these approaches is their dependence on dense 3D data input (*e.g.*, depth maps and point maps), which are often hard to acquire in certain real-world scenarios. Although estimating 3D attributes directly from images is possible [38, 42], it can introduce estimation errors and therefore degrade the performance, thus restricting their practical applicability. This naturally leads to the question: "*Can MLLMs understand the 3D world directly from videos without any explicit 3D data input?*"

Recent research [50] has shown that MLLMs face difficulties in understanding 3D geometry from encoded visual representations. This issue arises because these MLLMs process video frames as separate tokens through a visual encoder, which fails to capture crucial 3D geometric information, such as correspondences across frames [30]. Consequently, the MLLM backbone has to infer the

---

[*]denotes equal contribution

[♯]Corresponding Author

3D structure from the visual tokens to comprehend spatial relationships, which is both challenging and resource-intensive. This process often requires extensive supervision [6, 14, 54] and meticulous design to prevent issues such as catastrophic forgetting during fine-tuning [52]. These challenges highlight the critical need for methods that can incorporate 3D geometry priors into MLLMs.

In this work, we propose **V**ideo-3D **G**eometry **LLM** (VG LLM), a novel framework designed to explicitly integrate 3D visual geometry priors into MLLMs. To achieve this, we introduce a 3D visual geometry encoder that enriches input visual sequences with additional geometric information. Specifically, input images are processed by both a conventional visual encoder and the newly integrated 3D visual geometry encoder. The features extracted by these encoders are fused at the patch level and subsequently passed to the MLLM backbone. As the 3D visual geometry encoder is pre-trained on tasks such as point map prediction on pairs or sequences of images [38, 42, 44], it embeds strong 3D perception prior knowledge and is able to capture correspondences across frames. By doing so, VG LLM can effectively incorporate 3D geometry priors into the model and become more robust to viewpoint transformations, significantly improving its spatial reasoning abilities.

Extensive experiments have been conducted on various 3D scene understanding and spatial reasoning tasks, where the model accepts video input. These 3D scene understanding tasks include 3D visual grounding [7], 3D dense captioning [8], and 3D video object detection [45]. For spatial reasoning tasks, we evaluate our model on VSI-Bench [50] and CV-Bench [40]. The experiments show that our fine-tuned model outperforms larger spatial-enhanced models by a substantial margin. The results uncover several interesting findings: (1) Without explicit dense 3D inputs, our approach outperforms many leading 3D input-based models, underscoring its effective 3D geometric understanding. (2) By implicitly modeling inter-frame correspondences within the visual representation, our 8B model has learned strong egocentric-allocentric transformation capabilities, leading to significant improvements of precision by **11.9%** and F1 by **10.7%** on 3D video object detection. (3) On tasks that require complex spatial reasoning skills, *i.e.*, VSI-Bench [50], our 4B model attains an impressive average score of **47.3%**, surpassing even the best proprietary model, Gemini-1.5-Pro [39]. Furthermore, our 8B model sets a new state-of-the-art performance with a score of **50.7%**. This highlights the great utility of 3D geometry modeling in broader scenarios.

## 2 Related Work

**Multimodal Large Language Models (MLLMs).** MLLMs [2, 24, 26, 29, 32, 39, 40, 43] have achieved significant progress in 2D image and video understanding. However, recent findings [31, 50] indicate a critical limitation: current MLLMs still struggle with complex visual spatial reasoning tasks. To address this challenge, some research efforts [2, 40, 43] have focused on enhancing the models' ability to perceive and process spatial relationships through improved model representation. For example, Cambrian-1 [40] integrates self-supervised 2D visual representations with semantically rich features, aiming to provide a more comprehensive understanding of visual content that could benefit spatial reasoning. These methods focus on enriching the representation of each individual image, neglecting the 3D geometry information inherent in the continuous frames of a video. In contrast, our approach integrates 3D geometry priors from the video into the MLLM.

**3D Large Language Models.** Recent efforts [11, 13, 18, 20, 35, 58] have focused on enabling MLLMs to better understand 3D scenes. Previous work develops comprehensive 3D scene representations by using different types of 3D data. These include point cloud features [11, 13, 49], lifted multi-view image features [18, 20, 63], treating multi-view images as video sequences [35, 58]. The most closely related approaches are Video-3D LLM [58], which integrates positional information into the visual representation, and GPT4Scene [35], which constructs BEV maps through 3D reconstruction. In contrast to these methods, our model does not require any explicit dense 3D inputs.

**Spatial Reasoning.** Some work [5, 6, 14] has enhanced spatial understanding through large-scale synthesized VQA datasets for improved depth estimation. However, these methods primarily focus on static images, overlooking the dynamic and relational aspects inherent in complex spatial reasoning scenarios. To address this limitation, the Visual-Spatial Intelligence Benchmark (VSI-Bench) [50] was introduced, explicitly evaluating relational reasoning and egocentric-allocentric transformation abilities. Recognizing the scarcity of data for such complex scenarios, subsequent works like SAT [37] and SPAR [54] have proposed data generation pipelines to synthesize spatial QA datasets for supervised fine-tuning. In contrast, our model elicits 3D geometry information from the video and injects it into the model architecture, improving its spatial reasoning capabilties.

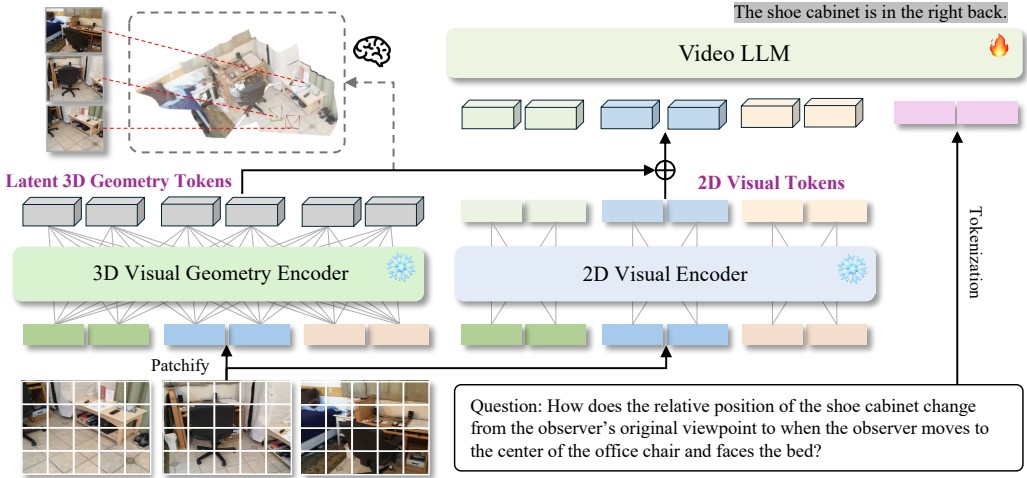

Figure 1: **The architecture of our VG LLM.** The 3D visual geometry encoder processes a sequence of images to produce globally geometry-aware visual features, while the 2D visual encoder extracts semantic-aware visual features from each individual image. 🧠 vividly shows that the latent 3D geometry tokens are able to recover the 3D scene if with a dense prediction head [42].

## 3 Method

We aim to model the 3D visual geometry in MLLMs without relying on explicit dense 3D input. To achieve this, as illustrated in Figure 1, we introduce a 3D visual geometry encoder to extract rich visual-geometric features from multiple images and feed them into the MLLM backbone. Section 3.1 outlines the architecture design of our model, and Section 3.2 details the training process.

### 3.1 Architecture

**Preliminary.** Given a sequence of RGB images $\{I_i\}_{i=1}^n$ and a natural language question $Q$, a conventional MLLM utilizes a 2D visual encoder to encode these images into image tokens $T_i^V \in \mathbb{R}^{\lfloor \frac{h}{p} \rfloor \times \lfloor \frac{w}{p} \rfloor \times c}$ first, where $I_i \in \mathbb{R}^{h \times w \times 3}$ and $p$ is the patch size. Then an MLLM backbone accepts $\{T_i^V\}_{i=1}^n$ as its input and outputs the response. In this work, we choose Qwen2.5-VL [2] as the MLLM backbone. Note that Qwen2.5-VL additionally compresses the image tokens to reduce the computational cost. Qwen2.5-VL groups spatially adjacent $2 \times 2$ patches into a single image token, resulting in a smaller set of image token input to MLLM backbone, $T_i^{V'} \in \mathbb{R}^{\lfloor \frac{h}{2p} \rfloor \times \lfloor \frac{w}{2p} \rfloor \times c}$.

**3D Visual Geometry Encoder.** To model 3D geometric information like inter-frame correspondences within input frames, we employ a 3D visual geometry encoder to extract such information. This 3D visual geometry encoder produces 3D visual geometry features $T_i^G \in \mathbb{R}^{\lfloor \frac{h}{p} \rfloor \times \lfloor \frac{w}{p} \rfloor \times c}$ from all input images $\{I_i\}_{i=1}^n$ jointly. 3D visual geometry models [44, 38, 42] are natural candidates for the 3D visual geometry encoder, as they are trained to capture inter-frame correspondences and reconstruct 3D scenes without relying on additional 3D priors. These 3D visual geometry models comprise three key components: an encoder for per-image feature extraction, a fusion decoder for cross-frame interaction, and task-specific prediction heads for 3D attributes. Since we focus on feature extraction, which embeds 3D geometry prior information, rather than directly outputting 3D attributes, we leverage the encoder and the fusion decoder as our 3D visual geometry encoder. Specifically, we choose VGGT [42] to extract 3D visual geometry features given its superior performance in 3D tasks.

**Visual Feature Fusion.** We fuse both the image tokens $\{T_i^{V'}\}_{i=1}^n$ and the 3D visual geometry features $\{T_i^G\}_{i=1}^n$ before passing them into the MLLM backbone. We first transform each $T_i^G$ into $T_i^{G'} \in \mathbb{R}^{\lfloor \frac{h}{2p} \rfloor \times \lfloor \frac{w}{2p} \rfloor \times c}$, which has the identical shape of $T_i^{V'}$, and then generate the geometry-augmented visual features $T_i^S = T_i^{G'} + T_i^{V'}$. The transformation of $T_i^G$ aligns with the spatial merging strategy in Qwen2.5-VL, where we concatenate spatially adjacent $2 \times 2$ features in $T_i^G$ and pass them to a two-layer MLP to output a single feature in $T_i^{G'}$.

To this end, the final visual features $\{T_i^S\}_{i=1}^n$ are concatenated with the text embeddings of the question $Q$ into the MLLM backbone to produce the response.

## 3.2 Training

Our VG LLM offers a versatile framework designed to integrate 3D vision priors into MLLMs for various 3D tasks. In this work, we demonstrate the application of VG LLM to 3D scene understanding and spatial reasoning tasks.

### 3.2.1 Applying VG LLM to 3D Scene Understanding

To validate the efficacy of the model in understanding and reasoning about 3D scenes, we adapt it to several 3D scene understanding tasks, *i.e.*, 3D visual grounding, 3D dense captioning, and 3D video object detection. In contrast to previous work on understanding 3D scenes, our model relies solely on RGB images as input during both training and inference, without any 3D scene data as prerequisites. For 3D dense captioning, while we utilize pre-detected 3D proposals as the input, our model itself operates solely on RGB images. We directly learn all 3D scene understanding tasks through text generation, which obviates the need for task-specific heads while maintaining a simple next-token prediction objective during training. The unified text generation objective allows us to mix all tasks into a combined dataset for multi-task training. The detailed data format can be found in the appendix.

**Coordinate System and Representation.** Since we do not utilize any ground truth information of the 3D scene, we follow VGGT [42] to employ the coordinate system of the first frame as the base coordinate system. All coordinates are transformed into this base coordinate system (except for 3D visual grounding, which represents the bounding box in each frame's coordinate system). All the numbers are expressed in plain text with 2 decimal places.

**3D Visual Grounding.** We follow previous work [54] to formulate the 3D visual grounding task as a 3D video grounding problem, which involves locating the frame index where the object appears and its 3D bounding box in the corresponding frame's coordinates in one feed-forward pass. Unlike SPAR [54], which generates axis-aligned boxes, our method directly predicts 3D-oriented bounding boxes, making it more suitable for real-world scenarios. Specifically, given the 3D scene represented by frames $\{I_1, I_2, \cdots, I_n\}$, and a natural language query $Q$, our model is designed to output both the frame index and the bounding box in the form $(x, y, z, w, h, d, \psi, \theta, \phi)$, where $(x, y, z)$ is the center coordinate, $(w, h, d)$ is the object size, and $(\psi, \theta, \phi)$ is the rotation angles.

**3D Dense Captioning.** We follow the recent work [58, 63] that decomposes the 3D dense captioning task into two phases, *i.e.*, detecting 3D object proposals with an off-the-shelf detector and generating object descriptions based on object coordinates. Specifically, given a frame sequence $\{I_1, I_2, \cdots, I_n\}$, we prompt our model to "describe the object located at $(x, y, z)$ in detail", where $(x, y, z)$ is the box center of the object to be described. This setup ensures a fair comparison with prior work and effectively evaluates the model's ability to understand positional and spatial relationships.

**3D Video Object Detection.** To investigate the capability of handling egocentric-allocentric transformation, we set up a 3D video object detection task based on the ScanNet [15]. Unlike previous benchmarks that focus on monocular or multi-view detection [45], our task requires models to detect all objects throughout the video in a unified coordinate system, without relying on explicit camera parameters or depth information. Since some objects aren't visible in the first frame, the model must track changes between frames, estimate camera movement, and convert object locations from a global view to the camera's perspective. Specifically, given continuous video frames $\{I_1, I_2, \cdots, I_n\}$, the model is tasked to detect all objects $\{(b_1, c_1), \cdots, (b_m, c_m)\}$ that appear in this video, where each $b_i$ in the form $(x, y, z, w, h, d, \psi, \theta, \phi)$ represents its bounding box in the unified coordinate system and $c_i$ is its category.

### 3.2.2 Instruction Tuning for Enhanced Spatial Reasoning.

Previous work [53, 62] has highlighted that existing pre-training and instruction tuning datasets often lack sufficient spatial-related phrases in their annotations, consequently hindering the spatial reasoning capabilities of MLLMs. To overcome this limitation, we leverage the SPAR-7M [54] dataset for instruction tuning. SPAR-7M is a comprehensive spatial reasoning dataset curated from three

richly annotated 3D datasets: ScanNet [15], ScanNet++ [51], and Structure3D [60]. It encompasses 33 diverse tasks, ranging from fundamental perception to mid-level viewpoint transformation and high-level scene imagination. As some work has shown that post-training on specific datasets can sometimes degrade the original performance on general benchmarks, we also incorporate a visual instruction tuning dataset, the LLaVA-Video-178K's LLaVA-Hound split [57], into our training pipeline to preserve the generalization capability.

## 4 Experiments

In this section, we first present the implementation details of our model, followed by an evaluation of our model's performance on 3D scene understanding tasks in Section 4.1, including 3D visual grounding, 3D dense captioning, and 3D multi-view detection. Next, we provide a comprehensive comparison with state-of-the-art methods on spatial reasoning benchmarks and generic multimodal benchmarks in Section 4.2. 3D scene understanding emphasizes 3D perception, while spatial reasoning focuses on interpreting and reasoning about spatial relationships in video. We train two models for 3D scene understanding and spatial reasoning tasks separately for fair comparison. Finally, we conduct a comprehensive analysis in Section 4.3 to reveal the effectivess of our model's core aspects: the integration of 3D geometry, the feature fusion strategy, and the data composition.

**Implementation and Training Details.** Our models are built upon two sizes of Qwen2.5-VL—3B and 7B [2], and integrated with VGGT-1B [42] as the 3D geometry encoder. We trained our models for one epoch on a mixed dataset, detailed as followed in this section, and employed Adam with a batch size of 64 and a warmup ratio of 0.03. During the warmup phase, the learning rate was gradually increased to 1e-5 before linearly decaying to 0. In each training step, a batch was randomly sampled from a single source from the mixed dataset. During training, the MLLM's visual encoder, the integrated 3D geometry encoder, and the multimodal connector are frozen, while the MLLM backbone remains unfrozen. All experiments were conducted on 8 H100 80G GPUs. For the 4B and 8B models, the training took 9 and 12 hours for 3D scene understanding, and 7 and 9 hours for spatial reasoning, respectively.

### 4.1 3D Scene Understanding Tasks

#### 4.1.1 Setting

**Datasets and Benchmarks.** To evaluate the versatility of our model across diverse 3D scene understanding tasks, we employ a multi-task learning approach on a combination of datasets.

- *3D Visual Grounding.* We leverage the ScanRefer [7] dataset, which provides 36,665 object descriptions paired with axis-aligned bounding boxes across 562 indoor scans. We follow SPAR [54] to reformulate 3D visual grounding as *3D spatial-temporal video grounding*, aiming to locate the target object's bounding box in camera coordinates along with its corresponding frame index. To determine the relevant appearance frame, we utilize the visible object annotations from Embodied-Scan [45]. We match the target 3D bounding box with those in EmbodiedScan based on their IoU and subsequently pick the optimal frame by comparing the 2D projection areas.
- *3D Dense Captioning.* We utilize the Scan2Cap benchmark [8] for 3D dense captioning, which requires generating descriptive captions for all objects within a scene. Following prior work [22, 58, 63], we use Mask3D-detected object proposals extracted by LEO [22] and task our model with generating captions conditioned on their center coordinates. To better leverage the visual geometry, we transform all object center coordinates to the coordinates of the first captured frame.
- *3D Video Object Detection.* For 3D video object detection, we curated a dataset from EmbodiedScan [45] consisting of consecutive frames and their corresponding visible object annotations in indoor scenes. Each sample comprises four consecutive frames sampled at 1 FPS, with all associated object instances transformed to the coordinate system of the initial frame. Consistent with EmbodiedScan, we divide the data into 958 training and 243 evaluation scenes. Within each scene, we randomly select 150/10 samples for training/evaluation.

**Comparison Baselines.** For the tasks of 3D visual grounding and dense captioning, our evaluation includes comparisons with both task-specific expert models and general-purpose 3D models. Specifically, for expert models, we compare our method against ScanRefer [7], MVT [23], and ViL3DRel [10] on ScanRefer [7]. On the Scan2Cap dataset [8], we include Scan2Cap [8], 3DJCG [4], D3Net [9], and Vote2Cap-DETR [12] for comparison. Furthermore, we compare our approach with the

generalist models, including Chat-3D v2 [21], Grounded 3D-LLM [13], LL3DA [11], LEO [22], LLaVA-3D [63], and Video-3D LLM [58]. For 3D video object detection, we compare our method with a baseline (Qwen2.5-VL-3B [2]) that does not incorporate 3D geometry information.

**Evaluation Metrics.** For ScanRefer [7], we report the accuracy at IoU thresholds of 0.25 and 0.5. For Scan2Cap [8], we use CIDEr (C), BLEU-4 (B-4), METEOR (M), and ROUGE (R) scores. The "@0.5" suffix indicates that these metrics are computed only for objects detected with an IoU of 0.5 or higher against the ground truth. Lastly, for 3D video object detection, we report the accuracy, recall, and F1 score for 20 common object classes at an IoU threshold of 0.25.

### 4.1.2 Results on 3D Visual Grounding

| Model | 3D Scene Input | Acc@0.25 | Acc@0.5 |
|---|---|---|---|
| ScanRefer [7] | ✓ | 37.3 | 24.3 |
| MVT [23] | ✓ | 40.8 | 33.3 |
| ViL3DRel [10] | ✓ | 47.9 | 37.7 |
| 3D-LLM [20] | ✓ | 30.3 | - |
| Chat-3D v2 [21] | ✓ | 35.9 | 30.4 |
| Grounded 3D-LLM [13] | ✓ | 47.9 | 44.1 |
| ChatScene [21] | ✓ | 55.5 | 50.2 |
| LLaVA-3D [63] | ✓ | 54.1 | 42.4 |
| Video-3D LLM [58] | ✓ | **58.1** | **51.7** |
| SPAR [54] | ✗ | 31.9 (48.8) | 12.4 (43.1) |
| VG LLM-4B (Ours) | ✗ | 36.4 (53.5) | 11.8 (47.5) |
| VG LLM-8B (Ours) | ✗ | **41.6** (57.6) | **14.9** (50.9) |

| Model | 3D Scene Input | C@0.5↑ | B-4@0.5↑ | M@0.5↑ | R@0.5↑ |
|---|---|---|---|---|---|
| Scan2Cap [8] | ✓ | 39.1 | 23.3 | 22.0 | 44.8 |
| 3DJCG [4] | ✓ | 49.5 | 51.0 | 24.2 | 50.8 |
| D3Net [9] | ✓ | 62.6 | 35.7 | 25.7 | 53.9 |
| Vote2Cap-DETR [12] | ✓ | 61.8 | 34.5 | 26.2 | 54.4 |
| LL3DA [11] | ✓ | 65.2 | 36.8 | 26.0 | 55.0 |
| Chat-3D-v2 [21] | ✓ | 63.9 | 31.8 | - | - |
| Grounded 3D-LLM [13] | ✓ | 70.2 | 35.0 | - | - |
| LEO [22] | ✓ | 72.4 | 38.2 | 27.9 | 58.1 |
| Chat-Scene [21] | ✓ | 77.1 | 36.5 | - | - |
| LLaVA-3D [63] | ✓ | 79.2 | **41.1** | **30.2** | **63.4** |
| Video-3D LLM [58] | ✓ | **80.0** | 40.2 | 28.5 | 61.7 |
| VG LLM-4B (Ours) | ✗ | 78.6 | 40.9 | 28.6 | 62.4 |
| VG LLM-8B (Ours) | ✗ | **80.0** | **41.5** | **28.9** | **62.6** |

Table 1: **The quantitative results on Scan-Refer.** The content in "()" indicates results with proposal refinement[3].

Table 2: **The performance on Scan2Cap.** VG LLM captions 3D object proposals using RGB cues only.

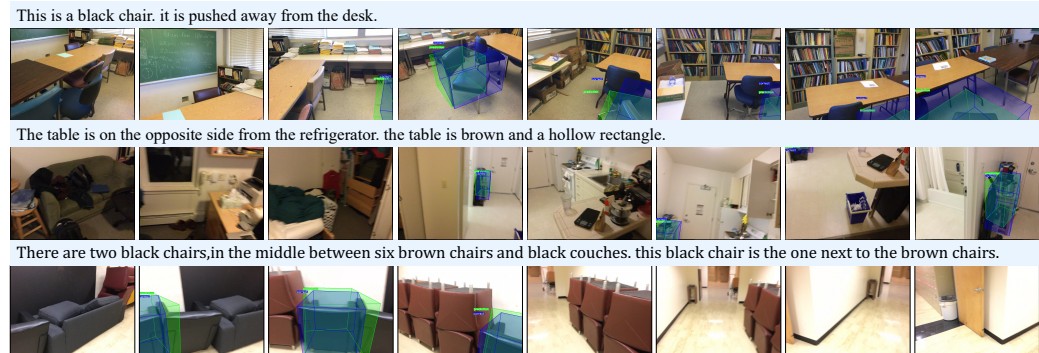

Figure 2: **Qualitative results for 3D visual grounding.** The ground truth and prediction are masked in blue and green, respectively. The predicted boxes are directly generated by our model without the refinement process.

In addition to the directly predicted bounding boxes, we also follow SPAR [54] to calculate the refined results with a proposal refinement process[3].

**VG LLM demonstrates strong 3D grounding capabilities from monocular RGB videos.** The performance on ScanRefer is shown in Table 1. Our 8B model achieves an accuracy of 41.6% at an IoU threshold of 0.25, significantly outperforming SPAR's 31.9% by 9.7 percentage points. Moreover, with the integration of a proposal refinement technique, Acc@0.25 increases substantially to 57.6%, making our results highly competitive with state-of-the-art methods like Video-3D LLM. This demonstrates that 3D visual grounding can be effectively approached in a video grounding manner.

**Qualitative results for 3D visual grounding.** Figure 2 illustrates the qualitative results on the 3D visual grounding task. For visualization purposes, the predicted and ground-truth 3D bounding boxes are projected onto their respective 2D image planes. Our method effectively handles spatial

---

[3]For proposal refinement, we compare the predicted box against all proposals detected by Mask3D and select the one with the highest IoU.

relationships between objects (*e.g.*, away, opposite, next to), accurately identifying the corresponding frame index and generating the 3D-oriented bounding box.

### 4.1.3 Results on 3D Dense Captioning

**Our model processes 3D object proposals generated by an off-the-shelf 3D object detector.** Once these objects are localized, the model generates descriptions using only the RGB images and the center coordinates of the detected objects, without requiring explicit depth or geometric data as additional inputs. This approach achieves a score of 80.0 C@0.5 and 41.5 B-4@0.5, comparable to previous state-of-the-art methods such as LLaVA-3D and Video-3D LLM. These results validate the effectiveness of our model for 3D dense captioning when conditioned on 3D proposals.

### 4.1.4 Results on 3D Video Object Detection

| Model | chair | cabinet | table | bin | couch | bed | bathtub | toilet | 20 Common Classes | | |
| --- | --- | --- | --- | --- | --- | --- | --- | --- | --- | --- | --- |
| | | | | | | | | | $P_{25}$ | $R_{25}$ | $F1_{25}$ |
| *4-Frame Setting* | | | | | | | | | | | |
| Qwen2.5-VL-3B | 37.7 | 10.2 | 35.0 | 23.1 | 39.0 | 64.8 | 32.4 | 68.8 | 32.6 | 27.9 | 30.0 |
| + Visual Geometry (**VG LLM-4B**) | 49.7 | 13.1 | 41.3 | 39.2 | 44.6 | 71.2 | 33.5 | 83.4 | 41.7 | 35.7 | 38.2 |
| △ *Improvement* | +12.0 | +2.9 | +6.3 | +16.1 | +5.6 | +6.4 | +1.1 | +14.6 | +9.1 | +7.8 | +8.2 |
| Qwen2.5-VL-7B | 41.2 | 11.6 | 36.5 | 30.2 | 41.1 | 68.2 | 36.6 | 68.7 | 34.6 | 31.0 | 32.5 |
| + Visual Geometry (**VG LLM-8B**) | 54.0 | 17.1 | 46.5 | 39.8 | 47.0 | 74.1 | 42.1 | 82.5 | 43.4 | 39.6 | 41.2 |
| △ *Improvement* | +12.8 | +5.5 | +10.0 | +9.7 | +5.9 | +5.9 | +5.5 | +13.8 | +8.8 | +8.6 | +8.7 |
| *6-Frame Setting* | | | | | | | | | | | |
| Qwen2.5-VL-3B | 32.8 | 7.8 | 31.3 | 20.9 | 32.2 | 58.8 | 36.5 | 66.1 | 27.8 | 24.1 | 25.7 |
| + Visual Geometry (**VG LLM-4B**) | 41.6 | 12.4 | 39.8 | 33.1 | 45.0 | 70.2 | 33.8 | 80.6 | 39.7 | 34.0 | 36.4 |
| △ *Improvement* | +8.8 | +4.6 | +8.5 | +12.2 | +12.8 | +11.4 | -2.7 | +14.5 | +11.9 | +9.9 | +10.7 |
| Qwen2.5-VL-7B | 36.1 | 10.6 | 32.7 | 25.0 | 40.7 | 64.6 | 38.4 | 68.6 | 31.8 | 28.0 | 29.6 |
| + Visual Geometry (**VG LLM-8B**) | 48.7 | 17.9 | 44.8 | 38.5 | 46.4 | 75.8 | 40.4 | 83.2 | 43.5 | 38.7 | 40.8 |
| △ *Improvement* | +12.6 | +7.3 | +12.1 | +13.5 | +5.7 | +11.2 | +2.0 | +14.6 | +11.7 | +10.7 | +11.2 |

Table 3: **The results on 3D video detection.** The reported object categories follow the monocular detection in EmbodiedScan, and we report the average F1 score per category.

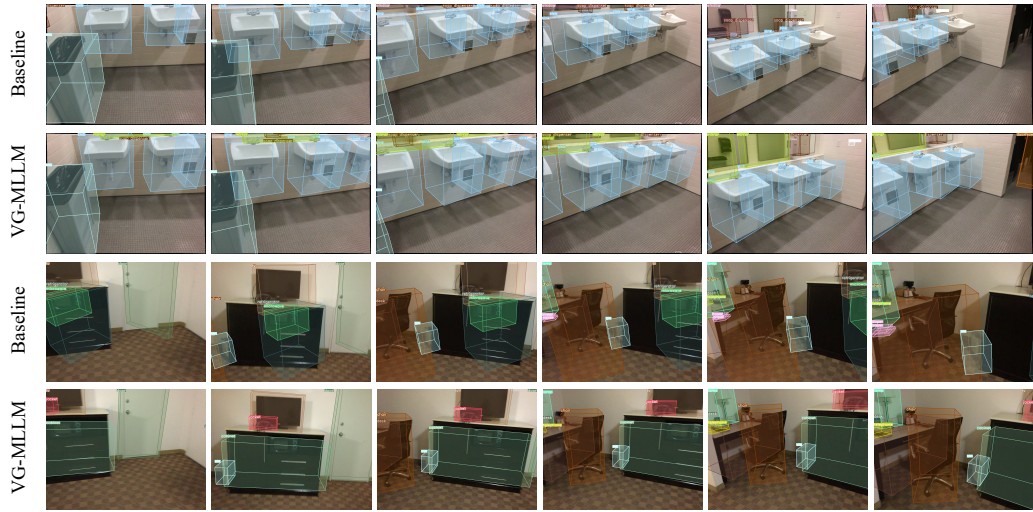

Figure 3: **Qualitative results for 3D video object detection.**

As shown in Table 3, we follow the monocular detection of EmbodiedScan [45] to evaluate the performance for categories that are common in daily life.

**3D geometry significantly boosts cross-frame detection performance.** Qwen2.5-VL [2], upon being fine-tuned on detection data, achieves promising performance on common object categories. Incorporating 3D geometry [42] yields a substantial improvement across all evaluation metrics. In the 4-frame setting, this approach elevates the average F1 score for the 4B model by 8.2% (from

30.0% to 38.2%) and for the 8B model by 8.7% (from 32.5% to 41.2%). This performance gain is attributed to the model's improved ability to comprehend the egocentric-allocentric transformation between frames through the utilization of 3D geometry. Figure 3 provides qualitative results on the 3D video object detection task.

**Integrating 3D geometry enhances the model's robustness to variations in frame count.** Table 3 shows that our method, although trained on 4-frame sequences, maintains strong performance when evaluated on longer sequences during inference. In contrast, the baseline's performance drops noticeably, underscoring the robustness of our approach to variations in the number of frames.

## 4.2 Spatial Reasoning Benchmarks

| Model | Avg. | Obj. Count | Abs. Dist. | Obj. Size | Room Size | Rel. Dist. | Rel. Dir. | Route Plan | Appr. Order |
|---|---|---|---|---|---|---|---|---|---|
| | | Numerical Answer | | | | Multiple-Choice Answer | | | |
| *Proprietary Models (API)* | | | | | | | | | |
| GPT-4o | 34.0 | 46.2 | 5.3 | 43.8 | 38.2 | 37.0 | 41.3 | 31.5 | 28.5 |
| Gemini-1.5-Flash | 42.1 | 49.8 | 30.8 | 53.5 | 54.4 | 37.7 | 41.0 | 31.5 | 37.8 |
| Gemini-1.5-Pro | 45.4 | 56.2 | 30.9 | 64.1 | 43.6 | 51.3 | 46.3 | 36.0 | 34.6 |
| *Open-source Models* | | | | | | | | | |
| InternVL2-8B | 34.6 | 23.1 | 28.7 | 48.2 | 39.8 | 36.7 | 30.7 | 29.9 | 39.6 |
| InternVL2-40B | 36.0 | 34.9 | 26.9 | 46.5 | 31.8 | 42.1 | 32.2 | 34.0 | 39.6 |
| LongVILA-8B | 21.6 | 29.1 | 9.1 | 16.7 | 0.0 | 29.6 | 30.7 | 32.5 | 25.5 |
| VILA-1.5-40B | 31.2 | 22.4 | 24.8 | 48.7 | 22.7 | 40.5 | 25.7 | 31.5 | 32.9 |
| LongVA-7B | 29.2 | 38.0 | 16.6 | 38.9 | 22.2 | 33.1 | 43.3 | 25.4 | 15.7 |
| LLaVA-NeXT-Video-72B | 40.9 | 48.9 | 22.8 | 57.4 | 35.3 | 42.4 | 36.7 | 35.0 | 48.6 |
| LLaVA-OneVision-72B | 40.2 | 43.5 | 23.9 | 57.6 | 37.5 | 42.5 | 39.9 | 32.5 | 44.6 |
| *Spatial-Enhanced Models* | | | | | | | | | |
| SAT-LLaVA-Video-7B | - | - | - | - | 47.3 | 41.1 | 37.1 | 36.1 | 40.4 |
| SPAR-8B | 41.1 | - | - | - | - | - | - | - | - |
| VG LLM-4B (Ours) | 47.3 | 66.0 | 37.8 | 55.2 | 59.2 | 44.6 | 45.6 | 33.5 | 36.4 |
| VG LLM-8B (Ours) | 50.7 | 67.9 | 37.7 | 58.6 | 62.0 | 46.6 | 40.7 | 32.4 | 59.2 |

Table 4: The comparison with state-of-the-art models on VSI-Bench. *Spatial-Enhanced Models* are models that are specialized for spatial reasoning.

### 4.2.1 Setting

**Datasets and Benchmarks.** To fully leverage the 3D knowledge inherent in the 3D visual geometry encoder, we train our model on a dataset sampled from SPAR-7M [54] and the LLaVA-Hound split of the LLaVA-Video-178K [57]. In our work, we sample only 234K and 63K data points from SPAR-7M and the LLaVA-Hound split of LLaVA-Video-178K, respectively, which constitute 3% and 25% of the original datasets.

We first evaluate our method on a video spatial reasoning benchmark, VSI-Bench [50], which evaluates the egocentric-allocentric transformation and relational reasoning capabilities of MLLMs. Then we test our model on an image spatial-related benchmark *i.e.*, CV-Bench [40]. CV-Bench [40] assesses 2D understanding through spatial relationships and object counting, while its 3D evaluation focuses on depth ordering and relative distance perception.

**Comparison Baselines.** We compare our model with state-of-the-art proprietary and open-source MLLMs (e.g., GPT-4o, Gemini-1.5-Pro, LLaVA-NeXT-Video) [1, 24, 26, 27, 32, 34, 39, 40, 46, 57]. We also include two spatial-enhanced MLLMs, SAT-LLaVA-Video-7B [37] and SPAR-8B [54], which are fine-tuned on datasets targeted for spatial reasoning.

**Evaluation Metrics.** For VSI-Bench, we adopt accuracy for multiple-choice tasks and Mean Relative Accuracy (MCA) for numerical tasks. MCA calculates the average accuracy across a range of confidence thresholds, where a prediction is considered correct if its relative error is within a specified threshold. For CV-Bench and other generic multimodal benchmarks, we report accuracy.

### 4.2.2 Results on Spatial Reasoning Tasks

**VG LLM achieves state-of-the-art performance on VSI-Bench.** A detailed comparison with leading models is presented in Table 4. As shown in the table, our 4B model achieves an impressive average accuracy of 47.3%, outperforming all competitors, including the leading proprietary model, Gemini-1.5 Pro. Futhermore, our 8B model sets a new state-of-the-art performance with an average accuracy of 50.7%. These results underscore the model's strong capability in understanding and reasoning from monocular videos.

| Model | 2D (%) | 3D. (%) | Avg. (%) |
|---|---|---|---|
| *Proprietary Models (API)* | | | |
| GPT-4V [34] | 64.3 | 73.8 | 69.1 |
| GPT-4o [24] | 74.8 | 83.0 | 78.9 |
| Gemini-1.5-Flash [39] | 70.9 | 71.8 | 71.4 |
| Gemini-1.5-Pro [39] | **77.1** | 77.6 | 77.3 |
| *Open-source Models* | | | |
| Mini-Gemini-HD-34B [27] | 71.5 | 79.2 | 75.4 |
| LLaVA-NeXT-34B [26] | 73.0 | 74.8 | 73.9 |
| Cambrian-1-34B [40] | 74.0 | 79.7 | 76.9 |
| SAT-LLaVA-Video-7B [37] | 73.0 | 83.8 | 78.4 |
| SPAR-8B [54] | 72.3 | 89.1 | 80.7 |
| VG LLM-4B (Ours) | 71.3 | 87.7 | 79.5 |
| VG LLM-8B (Ours) | 72.2 | **91.1** | **81.7** |

Table 5: **The comparison with state-of-the-art methods on CV-Bench.**

| Benchmark | Qwen2.5-VL-3B | VG LLM-4B | Qwen2.5-VL-7B | VG LLM-8B |
|---|---|---|---|---|
| Video-MME$_{w/o\ sub.}$ [17] | 60.1 | 57.4 | 62.9 | 59.3 |
| Video-MME$_{w/\ sub.}$ [17] | 60.9 | 59.8 | 61.1 | 63.9 |
| BLINK [19] | 46.6 | 48.9 | 55.9 | 51.5 |
| TempCompass$_{MC}$ | 62.2 | 63.9 | 71.8 | 67.8 |
| NextQA$_{MC}$ [48] | 77.3 | 74.8 | 81.4 | 79.3 |

Table 6: **Comparison of model performance on generic multimodal benchmarks.** To ensure a fair comparison, we evaluate Qwen2.5-VL with the resolution and sampling rate matched to those of VG LLM.

**Our model demonstrates generalization across distinct data sources.** To access the generalization capability of our method, we evaluate our approach on a spatial reasoning benchmark with different data sources, CV-Bench. This benchmark is constructed by repurposing the traditional CV datasets (*i.e.*, ADE20K, COCO, and Omni3D) to a vision-centric MLLM benchmark. As illustrated in Table 5, our method achieves the highest accuracy on 3D tasks at 91.1%. These results demonstrate our model can also generalize well across the spatial reasoning benchmarks with out-of-the-domain data sources.

### 4.2.3 Results on Generic Multimodal Benchmarks

**Enhancing spatial understanding incurs negligible loss on general multimodal performance.** Our model integrates 3D geometry information and is fine-tuned on spatial reasoning datasets to bolster its spatial understanding capabilities. As presented in Table 6, these enhancements slightly compromise VG LLM's performance on general multimodal benchmarks when compared to the baseline model. VG LLM-4B even achieves improvements on BLNK (+2.3%) and TempCompass$_{MC}$ (+1.7%). This suggests that augmenting spatial reasoning capabilities can not only preserve but also enhance performance on specific multimodal tasks.

## 4.3 Analysis

### 4.3.1 3D Scene Understanding

| Type | ScanRefer | | Scan2Cap | | 3D Video Detection | | |
|---|---|---|---|---|---|---|---|
| | Acc@0.25 | Acc@0.5 | CIDEr@0.5 | BLEU-4@0.5 | $P_{25}$ | $R_{25}$ | $F1_{25}$ |
| No Additional Info. (Baseline) | 31.9 (49.9) | 9.3 (43.8) | 58.0 | 36.3 | 32.6 | 27.9 | 30.0 |
| Cross-Attn (1 Layer) | 33.7 (51.3) | 10.7 (45.2) | 74.7 | 40.1 | 38.0 | 33.6 | 35.4 |
| Cross-Attn (3 Layers) | 34.4 (51.3) | 10.5 (45.2) | 75.7 | 40.2 | 38.5 | 44.0 | 35.4 |
| Concat+MLP | 27.7 (47.0) | 6.8 (41.3) | 75.7 | 39.9 | 37.1 | 32.5 | 34.4 |
| Add (Ours) | **36.4 (53.5)** | **11.8 (47.5)** | **78.6** | **40.9** | **41.7** | **35.7** | **38.2** |
| Pred Camera Info. | 32.1 (50.0) | 9.9 (43.9) | 56.8 | 36.3 | 33.3 | 28.1 | 30.3 |
| Pred Depth Info. | 32.3 (49.7) | 9.7 (43.7) | 57.1 | 36.1 | 32.1 | 27.3 | 29.3 |
| Pred Point Info. | 31.7 (49.7) | 9.6 (43.8) | 67.7 | 38.2 | 33.0 | 28.5 | 30.4 |
| Pred (Depth + Camera) Info. | 32.6 (49.8) | 9.8 (43.7) | 58.2 | 36.6 | 31.7 | 27.1 | 29.1 |
| Latent 3D Geometry (Ours) | **36.4 (53.5)** | **11.8 (47.5)** | **78.6** | **40.9** | **41.7** | **35.7** | **38.2** |

Table 7: **Ablation study of the effects of 3D visual geometry modeling.** All models are fine-tuned using the same training data and built upon Qwen2.5-VL-3B.

In this section, we conduct a further analysis on the feature fusion strategies and the types of added signals. To investigate the effect of feature fusion strategies, we have experimented with several feature fusion strategies, including 1) "Cross-Attn", which consists of multiple blocks of cross-attention modules and MLPs with skip connections. 2D visual tokens serve as queries, while 3D visual tokens serve as keys and values. 2D positional embeddings are added to both the queries and keys; 2) "Concat+MLP" first concatenates the 2D and 3D visual tokens along the feature dimension, followed by an MLP to transform them into the text embedding space; 3) "Add" is the strategy employed in our paper, which directly adds the 2D and 3D visual tokens at a patch level.

Besides the visual geometry feature adopted in our model, VGGT can also directly predict the camera poses (*Pred Camera Info.*), the depth maps (*Pred Depth Info.*), and the point maps (*Pred Point Info.*) of each image frame. For these predicted spatial signals, we adopt a two-layer MLP to transform them and add to the corresponding 2D visual token. All models are trained and evaluated following the datasets and benchmarks in Section 4.1.

**Directly adding the geometry features obtains the best results among all compared feature fusion strategies.** As Table 7 illustrates, the "Cross-Attn" strategy significantly improves performance over the baseline. Moreover, increasing the number of layers to three yields even greater performance gains. While "Concat+MLP" performs worse than the baseline on ScanRefer, it outperforms it on Scan2Cap and 3D video object detection. Nevertheless, "Add" surpasses all other comparison methods despite its simplicity.

**Incorporating point maps enhances 3D scene understanding, while camera or depth information offers no clear benefits.** After fine-tuning on 3D downstream tasks, the baseline model Qwen2.5-VL shows enhanced 3D scene understanding. However, it continues to struggle with accurately localizing and describing 3D objects, as well as learning spatial transformations across different frames. As illustrated in Table 7, incorporating predicted camera or depth information alone provides no clear benefit. In contrast, the inclusion of predicted point information demonstrates a significant advantage for the 3D dense captioning task, boosting the CIDEr@0.5 score on Scan2Cap from 58.0 to 67.7 and the BLEU-4@0.5 from 36.3 to 38.2. It also provides a marginal improvement in 3D video object detection. These results indicate that current MLLMs still lack adequate spatial modeling in both data curation and architectural design.

**Visual geometry features are more helpful than predicted spatial information.** Although the point maps predicted directly by VGGT can enhance the spatial understanding ability of Qwen2.5-VL, these signals are relatively sparse and may contain errors. In contrast, directly using the visual geometry features provided by VGGT not only incorporates these spatial signals simultaneously but also reduces the impact of noise on performance, thereby achieving better results.

### 4.3.2 Spatial Reasoning

| # | Data | Model | VSI-Bench | | | | | | | | | CV-Bench (3D) | | |
|---|------|-------|-----------|---|---|---|---|---|---|---|---|---------------|---|---|
| | | | Obj. Count | Abs. Dist. | Obj. Size | Room Size | Rel. Dist. | Rel. Dir. | Route Plan | Appr. Order | Avg | Depth | Distance | Avg |
| 1 | ∅ | Qwen2.5-VL-7B | 25.4 | 10.4 | 36.4 | 29.1 | 38.2 | 37.9 | 30.9 | 27.5 | 29.5 | 84.5 | 76.5 | 80.5 |
| 2 | S1 | Qwen2.5-VL-7B | **68.6** | 36.0 | 57.9 | 59.7 | 45.8 | 38.9 | 30.4 | **60.2** | 49.8 | 87.0 | 87.2 | 87.1 |
| 3 | | VG LLM-8B | 67.9 | **37.7** | **58.6** | **62.0** | **46.6** | **40.7** | **32.4** | 59.2 | **50.7** | **92.3** | **89.8** | **91.1** |
| 4 | S2 | Qwen2.5-VL-7B | 71.3 | 48.3 | 68.5 | **65.5** | 65.1 | 77.5 | 40.2 | 16.7 | 56.6 | **66.8** | **74.5** | **70.7** |
| 5 | | VG LLM-8B | **71.7** | **53.8** | **68.8** | 62.1 | 63.8 | **83.0** | **44.3** | **18.4** | **58.2** | **66.8** | 73.3 | 70.1 |
| 6 | S1+S2 | Qwen2.5-VL-7B | 70.2 | 51.0 | **69.8** | 64.4 | 67.0 | 79.1 | 45.4 | 31.6 | 59.8 | 87.0 | 83.8 | 85.4 |
| 7 | | VG LLM-8B | **71.4** | **56.8** | 69.0 | **69.1** | **67.9** | **83.2** | **47.4** | **32.5** | **62.2** | **92.0** | **89.5** | **90.8** |

Table 8: **Ablation study on spatial reasoning tasks.** ∅ indicates testing in a zero-shot setting. S1 is the training data detailed in Section 4.2.1, and S2 refers to the VLM-3R dataset.

We then investigate the effects of fine-tuning and visual geometry in Table 8. The table illustrates that fine-tuning on our mixed dataset (S1) leads to a substantial performance gain in measurement estimation (rows 1 vs. 2). In addition to the data used in this paper (S1), the VLM-3R dataset [16] (S2), which is specially curated for the task types in VSI-Bench, is also incorporated in our analysis. After incorporating the 3D geometry into the model architecture, the results show consistent gains with the different data compositions, *i.e.*, S1, S2 and S1+S2.

## 5 Conclusion

We present a novel framework to enhance MLLMs' 3D spatial understanding capability, which incorporates a 3D visual geometry encoder to provide latent 3D geometric information given only video inputs. While being straightforward, our extensive experiments show that our model can outperform larger spatial-enhanced models on various 3D scene understanding tasks and spatial reasoning benchmarks, without relying on any explicit 3D scene input.

**Acknowledgements** This work is supported by National Key R&D Program of China (Project No. 2022ZD0161200, 2022ZD0161201). This work is also supported by Hong Kong Research Grant Council - Early Career Scheme (Grant No. 24200223).

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

# A    Experimental Setting and Details

## A.1    Evaluation Details.

We utilize the LMMs-Eval [56] to evaluate our method. LMMs-Eval offers flexible functionalities that support the customization of specific tasks and model interfaces, thereby standardizing the evaluation process of MLLMs. For inference, we employed greedy sampling to generate model outputs.

**Spatial Reasoning and General Multi-modal Benchmarks.**    For benchmarks already integrated into LMMs-Eval, *e.g.*, VSI-Bench [50], Video-MME [17], and TempCompass [33], we utilize their original default configurations. For benchmarks not yet implemented in LMMs-Eval, *i.e.*, BLINK [19] and CV-Bench [40], we customize the task configurations and metrics according to their official implementations. For each input video in video-based benchmarks, we uniformly sample 32 frames from the entire duration.

**3D Visual Grounding.**    For 3D visual grounding, the model is tasked with locating the frame index where the target object appears and the oriented bounding box within its coordinate system. Given a predicted frame index and bounding box, we first transform the bounding box to the world coordinate system. Then, the ground truth bounding box is extended to an oriented bounding box by appending (0, 0, 0) to represent its orientation. Finally, we calculate the Intersection-over-Union (IoU) between the predicted and ground truth bounding boxes. In the proposal refinement process, we utilize the Mask3D-detected object proposals extracted by LEO [22], and select the proposal that has the highest IoU with the predicted box.

**3D Dense Captioning.**    For 3D dense captioning, we follow previous work [63, 58] to decompose the task into two stages, *i.e.*, detecting all object proposals and generating object descriptions based on object center coordinates. Specifically, we first leverage the Mask3D-detected object proposals extracted from LEO [22] to obtain the box's center coordinates within the coordinate system of the reference frame. This can be achieved by multiplying the coordinates by the extrinsic matrix [45]. Then, the model is asked to generate descriptions based on the coordinates through greedy sampling. Lastly, we calculate the CIDEr, BLEU, Rouge, and METEOR scores utilizing the COCO caption evaluation toolkit[4].

**3D Video Object Detection.**    For each predicted bounding box, we find its best match among the unused ground truth boxes of the same category through greedy matching. Specifically, we calculate the IoU with all unused ground truth boxes of that category and select the one with the highest IoU if it exceeds a threshold (e.g., 0.25). Once a ground truth is matched, it is marked as used and cannot be matched again. Finally, for each category, the precision, recall, and F1 score are calculated based on the number of true positives, false positives, and false negatives.

Additionally, we provide the prompt for 3D scene understanding tasks in Table 9.

# B    Data preparation

All datasets used in this research are publicly available, and we will provide the details for the data preparation in this section.

**SPAR-7M.**    We follow the official codebase[5] to mix data and draw visual markers on the input images. Since the navigation type contains images of varying lengths, we discard annotations of this type for simplicity.

**LLaVA-Video-178K (LLaVA-Hound Split).**    For each input video, we sample frames with a sampling rate of 2 FPS, while constraining the total frame count between 4 and 8 through adaptive sampling.

**3D Scene Understanding.**    For both ScanRefer and Scan2Cap, we uniformly sample 32 frames for each scene. For 3D video object detection, each entry in the training set contains 4 consecutive frames sampled at 1 FPS.

---

[4]https://github.com/tylin/coco-caption
[5]https://github.com/fudan-zvg/spar

*3D Visual Grounding*
$\mathbf{X}_{\texttt{User}}$: Frame-0: `<image>`Frame-1:`<image>`Frame-2: `<image>`Frame-3: `<image>`···
Localize the first clear frame in the video showing the object described in the text.
Text: There is a beige wooden bookshelf. placed next to another bookshelf.
Output a JSON dictionary with the frame index in "frame" and its 3D bounding box in "box_3d" in the frame's coordinates.
$\mathbf{X}_{\texttt{Agent}}$:
```json
{"frame": 12, "bbox_3d": [-0.63, -0.83, 2.43, 3.0, 0.59, 2.35, -2.32, 1.18, 3.05]}
```

*3D Dense Captioning*
$\mathbf{X}_{\texttt{User}}$: `<image><image><image><image>`···
Carefully watch the video and describe the object located at [-0.89, -0.74, 2.45] in detail.
$\mathbf{X}_{\texttt{Agent}}$: A white cabinet in the corner of the room. in the direction from the door and from the inside. it will be on the left, there is a small brown table on the left side of the cabinet and a smaller table on the right side of the cabinet.

*3D Video Object Detection*
$\mathbf{X}_{\texttt{User}}$: `<image><image><image><image>`···
Detect the 3D bounding boxes in the camera coordinate system of the first frame.
Output a json list where each entry contains the object name in "label" and its 3D bounding box in "box_3d".
The 3D bounding box format should be [x_center, y_center, z_center, x_size, y_size, z_size, yaw, pitch, rolll].
$\mathbf{X}_{\texttt{Agent}}$:
```json
[
    {"label": "bag", "bbox_3d": [0.0, -0.3, 1.0, 0.26, 0.26, 0.15, 1.67, 0.96, -2.98]},
    ...
]
```

Table 9: **The prompt for 3D scene understanding tasks.** For 3D visual grounding and 3D video object detection, the output should be in JSON format, while for 3D dense captioning, it should be a natural language description.

## C  Detailed Results on 3D Video Object Detection

Detailed quantitative results for our 3D video object detection are presented in Table 10 and 11. As illustrated in the table, our method significantly outperforms the baseline that does not utilize visual geometry across most categories. Nevertheless, we observe that detecting small objects such as pillows, lamps, and backpacks remains challenging.

## D  More Visualization

Figure 4 shows more visualization results for 3D visual grounding. The first three examples are positive cases where our model successfully locates the 3D bounding boxes for different categories like chair, cabinet, and document organizer. While the model correctly identifies these objects, we notice that it still struggles with accurately predicting their orientation. The last two examples are negative cases. While the predictions might appear correct in the 2D projection, the model's inaccurate depth estimation causes some discrepancies in 3D space.

We provide more visualization results on 3D video object detection in Figure 5. From this figure, we observe that while the baseline performs decently in detecting objects in a given video, incorporating 3D geometry significantly improves both detection precision and recall. For instance, in the first example, the baseline's prediction for the desk mismatches the ground truth, whereas our approach improves the prediction box of the desk.

| Object | Qwen2.5-VL-3B | VG LLM-4B | Qwen2.5-VL-7B | VG LLM-8B |
|---|---|---|---|---|
| chair | 32.8 | 41.6 | 36.1 | 48.7 |
| pillow | 11.4 | 15.9 | 12.4 | 18.9 |
| cabinet | 7.8 | 12.4 | 10.6 | 17.9 |
| table | 31.3 | 39.8 | 32.7 | 44.8 |
| lamp | 5.6 | 15.6 | 8.0 | 15.9 |
| couch | 32.2 | 45.0 | 40.7 | 46.4 |
| desk | 31.6 | 38.5 | 34.4 | 32.3 |
| stand | 20.6 | 41.4 | 31.3 | 47.3 |
| bed | 58.8 | 70.2 | 64.6 | 75.8 |
| backpack | 28.9 | 37.5 | 28.6 | 40.4 |
| bathtub | 36.5 | 33.8 | 38.4 | 45.2 |
| ottoman | 0.0 | 2.8 | 2.7 | 5.3 |
| dresser | 31.2 | 41.1 | 36.4 | 44.5 |
| bin | 20.9 | 33.1 | 25.0 | 38.5 |
| toilet | 66.1 | 80.6 | 68.6 | 83.2 |
| refrigerator | 28.9 | 64.3 | 33.2 | 62.0 |
| stove | 37.8 | 68.5 | 55.9 | 69.5 |
| microwave | 11.5 | 21.1 | 7.8 | 25.8 |
| monitor | 15.5 | 22.1 | 20.3 | 31.7 |
| computer | 4.4 | 3.1 | 5.2 | 11.6 |

Table 10: **The detailed results on 3D video object detection in 6-frame setting**. We report the F1 score for all categories.

| Object | Qwen2.5-VL-3B | VG LLM-4B | Qwen2.5-VL-7B | VG LLM-8B |
|---|---|---|---|---|
| chair | 37.7 | 49.7 | 41.2 | 54.0 |
| pillow | 11.9 | 17.9 | 13.2 | 22.9 |
| cabinet | 10.2 | 13.1 | 11.6 | 17.1 |
| table | 35.0 | 41.3 | 36.5 | 46.5 |
| lamp | 4.4 | 16.4 | 9.0 | 18.9 |
| couch | 39.0 | 44.6 | 41.1 | 47.0 |
| desk | 33.7 | 39.0 | 34.5 | 41.0 |
| stand | 23.3 | 37.9 | 33.7 | 46.1 |
| bed | 64.8 | 71.2 | 68.2 | 74.1 |
| backpack | 27.4 | 42.0 | 33.3 | 41.4 |
| bathtub | 32.4 | 33.5 | 36.6 | 42.1 |
| ottoman | 3.2 | 3.8 | 0.0 | 0.0 |
| dresser | 33.1 | 40.3 | 38.2 | 49.2 |
| bin | 23.1 | 39.2 | 30.2 | 39.8 |
| toilet | 68.8 | 83.4 | 68.7 | 82.5 |
| refrigerator | 32.2 | 57.3 | 38.5 | 53.6 |
| stove | 62.5 | 68.8 | 61.2 | 68.6 |
| microwave | 25.9 | 32.1 | 21.3 | 33.3 |
| monitor | 22.3 | 29.1 | 21.3 | 34.0 |
| computer | 6.8 | 3.8 | 12.0 | 11.4 |

Table 11: **The detailed results on 3D video object detection in 4-frame setting**. We report the F1 score for all categories.

# E   Limitations

While our model is built upon 3B and 7B MLLM backbones, its capabilities remain constrained by the model's inherent capacity. Scaling up the model size could potentially enhance both fundamental abilities and generalization performance. Furthermore, this work focuses exclusively on supervised fine-tuning, whereas emerging research demonstrates that reinforcement learning can significantly enhance model reasoning capabilities. Since VGGT is trained on commonly used 3D scene datasets, evaluating it on benchmarks built from that same data (such as ScanNet) poses a greater risk of overestimating its performance compared to evaluating it on out-of-distribution benchmarks. We leave these for future work.

# F    Broader Impact

VG LLM is built upon Qwen2.5-VL and consequently shares many common challenges associated with multi-modal large language models (MLLMs). These challenges include hallucinations in visual understanding, inherited biases from the base models, and susceptibility to adversarial inputs. Despite these limitations, releasing VG LLM to the research community would be highly beneficial, as it could catalyze further advancements in 3D world understanding by leveraging Video LLMs and 3D vision geometry priors.

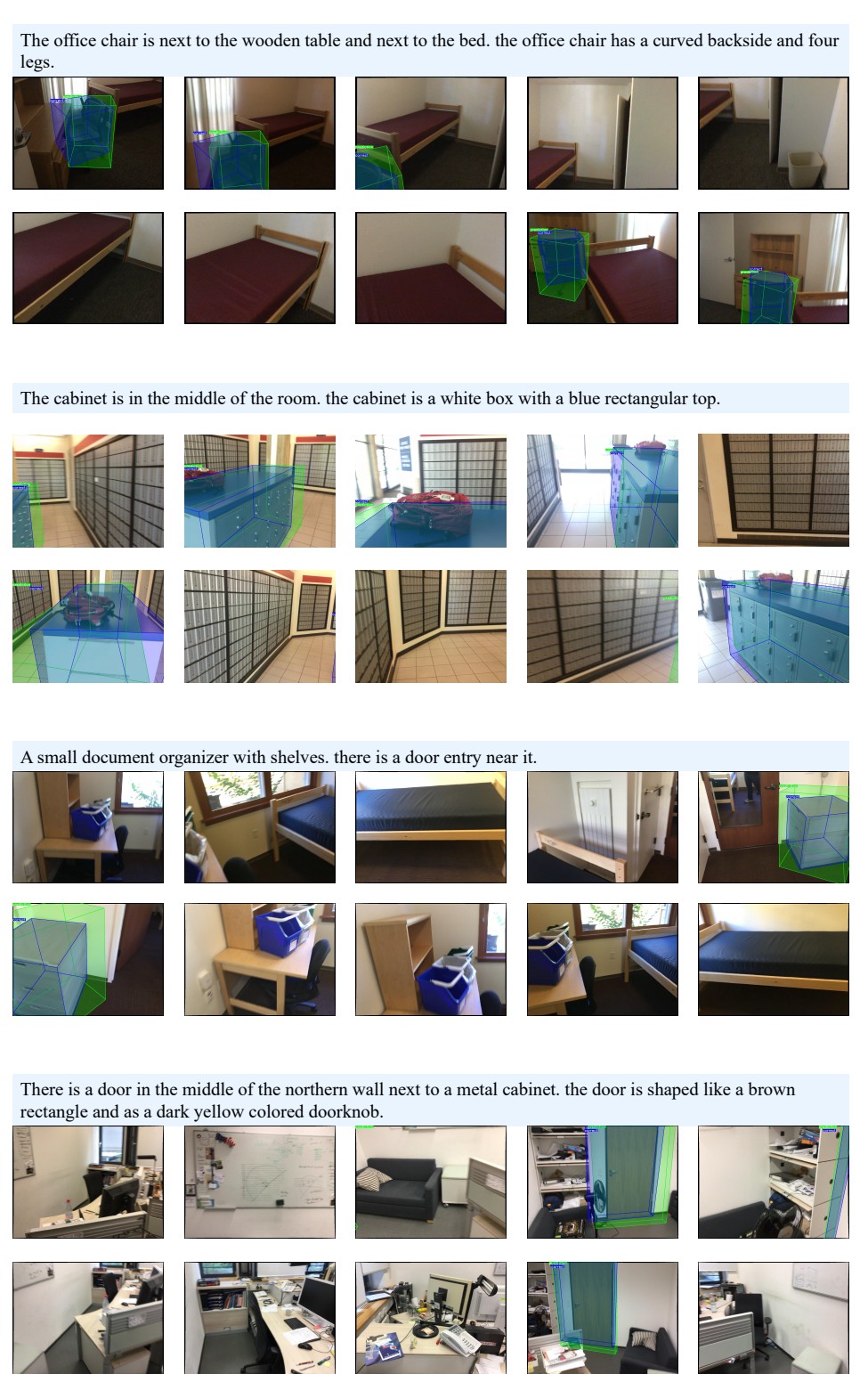

Figure 4: **Visualization of 3D visual grounding.** The first three examples are positive, whereas the last two are negative cases.

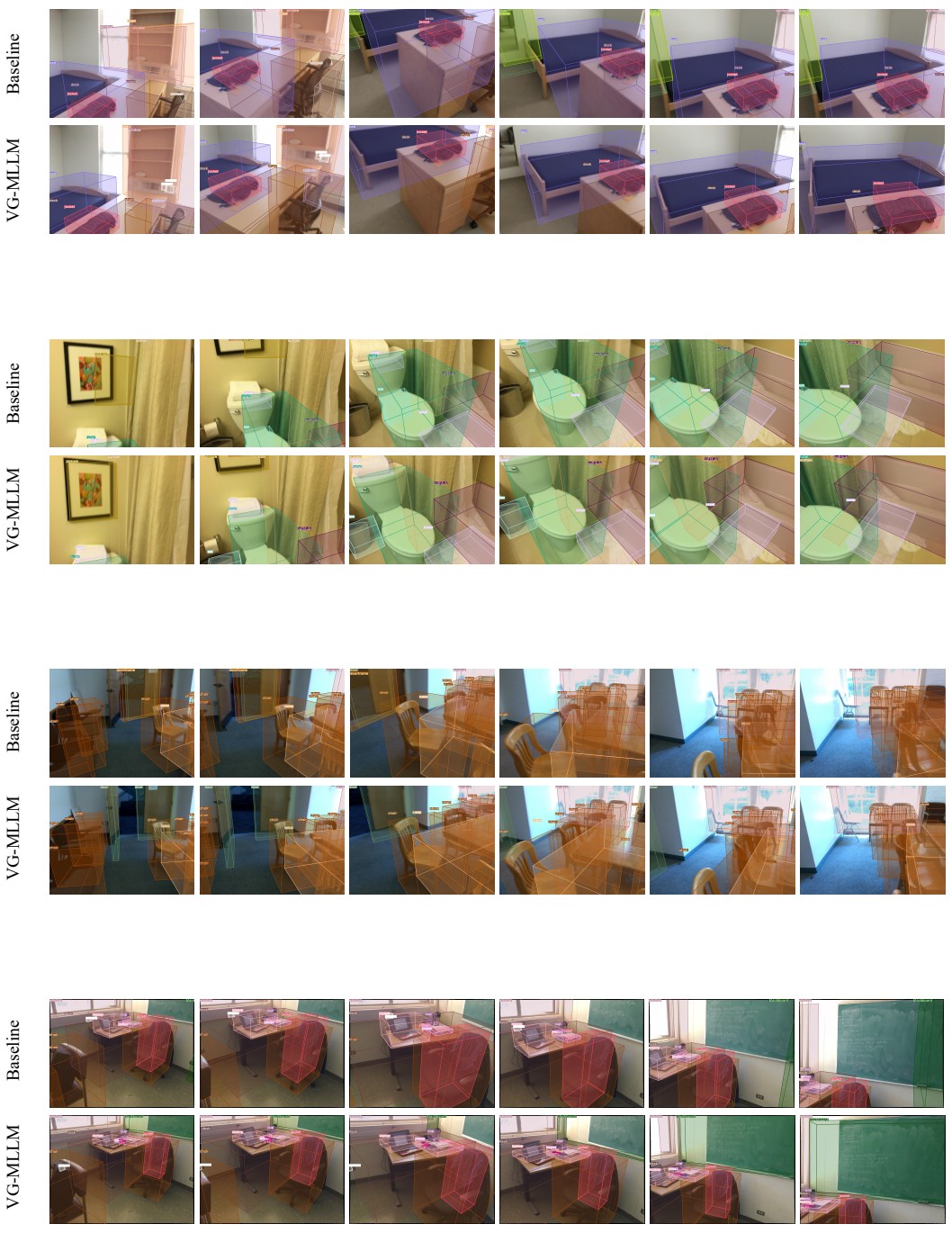

Figure 5: **Visualization of 3D video object detection results.**

