# OpenReview forum: "Learning from Videos for 3D World: Enhancing MLLMs with 3D Vision Geometry Priors"
_NeurIPS.cc/2025/Conference — NeurIPS 2025 poster_

### Official Review · Reviewer_FeDL · 2025-06-15

**Clarity:** 3
**Significance:** 3
**Originality:** 3
**Rating:** 5
**Confidence:** 3

**Summary:**

This paper presents VG LLM (Video-3D Geometry LLM), a novel multimodal large language model (MLLM) framework designed to enhance spatial understanding from videos without relying on explicit dense 3D inputs. The key idea is to leverage a pretrained 3D visual geometry encoder (VGGT) to extract latent 3D geometric priors from video frames. The proposed model is evaluated on a variety of 3D vision-language tasks, including 3D visual grounding, dense captioning, video object detection, and spatial reasoning benchmarks. VG LLM achieves competitive or even state-of-the-art performance compared to models that rely on explicit 3D inputs, demonstrating the effectiveness.

**Questions:**

Please refer to the above weaknesses.

Other minor questions:
1. Would VG LLM fail under low-quality videos where the pretrained VGGT may generalize poorly?

**Ethical Concerns:**

["NO or VERY MINOR ethics concerns only"]

**Final Justification:**

The idea of injecting the knowledge of a foundation 3D reconstruction model (VGGT) into MLLMs is promising, and I think the experiments have demonstrated its effectiveness. Thus, I tend to accept this paper.

**Limitations:**

Yes, in the appendix.

**Paper Formatting Concerns:**

None.

**Quality:**

3

**Strengths And Weaknesses:**

Strengths:
1. VG LLM does not require explicit dense 3D inputs, making the model more flexible and generalizable.
2. The performance of VG LLM is superior.
3. The paper is well-written and easy to follow.

Weaknesses:
1. Although VGGT demonstrates excellent performance for multi-view 3D reconstruction, its output is in a relative scale rather than metric. How does VG LLM handle this scale ambiguity without using any explicit 3D inputs?
2. The metrics in Table 2 (C@0.5, B-4@0.5, etc.) should be explained in more detail.

---

> ### Author Rebuttal · Authors · 2025-07-31
>
> Thanks for your valuable comments!
>
> * **(W1) How does our method handle the scale ambiguity?**
>
>     While VGGT indeed provides features at a relative scale, our model effectively learns the absolute scale of objects in the real-world scenes by fine-tuning on customized datasets. Specifically, we use a training dataset encompassing tasks such as 3D visual grounding, 3D dense captioning, and 3D video object detection for 3D scene understanding. The 3D coordinates in these tasks embed the absolute scale information about common objects in the real-world scenes. Our model can learn to infer the scale of unseen objects within the scene by leveraging the relative scale provided by VGGT and the absolute scale of seen objects learned during training.
>
> * **(W2) The explanations for C@0.5 and B-4@0.5.**
>
>     We follow the previous work [1,2] to decompose the 3D dense captioning task into two stages, i.e., detecting 3D object proposals with an off-the-shelf detector and generating object descriptions based on detected object coordinates. The evaluation metrics C@0.5, B−4@0.5, M@0.5, and R@0.5 represent the CIDEr, BLEU-4, METEOR, and ROUGE scores, respectively. The @0.5 suffix specifies that these metrics are evaluated only on detected objects with an IoU ≥0.5 against a ground-truth object.
>
> * **(Q1) The influence of video quality.**
>
>     To investigate the impact of the low-quality videos that VGGT may generalize poorly, we conducted experiments on 3D dense captioning. Specifically, we downsampled each video frame by reducing its width to a certain pixel value (maintaining aspect ratio) and then resized it back to the original dimension before feeding it as input for VGGT. Note that the input resolution for the ViT within the MLLM was not adjusted for fair comparison.
>
>     As shown in the table, the performance degrades as the downsampling width decreases from 518 to 224, indicating an impact from accumulated errors from VGGT. However, this degradation is limited, highlighting the robustness of our methodology.
>
>     | width | CIDER@0.25 | B-4@0.25 |
>     |:-----:|:----------:|:--------:|
>     | 224px |    73.4    |   40.0   |
>     | 336px |    74.9    |   40.1   |
>     | 448px |    76.4    |   40.5   |
>     | 518px |    78.6    |   40.9   |
>
> [1] Chat-Scene: Bridging 3D Scene and Large Language Models with Object Identifiers. Huang et al. NeurIPS 2024.
>
> [2] An Embodied Generalist Agent in 3D World. Huang et al. ICML 2024.

---

> > ### Comment · Reviewer_FeDL · 2025-08-05
> >
> > Thanks to the authors for their rebuttal. I have read all the reviewers' questions and the response. The authors have addressed my concerns. Although I think learning the absolute scale by fine-tuning on customized datasets may not generalize well to other datasets or domains, I believe the idea of injecting the knowledge of a foundation 3D reconstruction model (VGGT) into MLLMs is promising. Overall, I tend to accept this paper.

---

> > > ### Author Response · Authors · 2025-08-05
> > >
> > > Thanks for your positive recommendation and the supportive feedback on our paper. We are glad that our responses have addressed your concerns.

---

### Official Review · Reviewer_MDYF · 2025-06-22

**Clarity:** 3
**Significance:** 3
**Originality:** 3
**Rating:** 4
**Confidence:** 5

**Summary:**

The key innovation of this work is to learn 3D knowledge directly from videos by integrating implicit 3D geometry priors. The proposed VG LLM architecture uses two parallel encoders: a standard 2D visual encoder to extract semantic features from individual frames, and a dedicated 3D visual geometry encoder (instantiated with VGGT) that processes the video sequence to extract latent geometry-aware features. These two sets of features are fused and fed into the MLLM backbone. The authors conduct extensive experiments on a wide range of tasks, including 3D scene understanding (visual grounding, dense captioning, video object detection) and complex spatial reasoning (VSI-Bench, CV-Bench, BLINK). Notably, their 4B parameter model achieves state-of-the-art results on the challenging VSI-Bench, outperforming even proprietary models like Gemini-1.5-Pro.

**Questions:**

Questions
1. Have you considered or experimented with other 3D visual geometry encoders (e.g., Dust3R)? Would the framework still be effective with a less powerful or different type of geometry encoder, or is the success fundamentally linked to the specific capabilities of VGGT?

2. Could you please provide more details on the computational overhead at inference time? Specifically, what is the increase in latency and memory usage for a typical video input compared to the Qwen2.5-VL baseline?

3. The paper focuses on a 4B model. What is your hypothesis on how the performance gains from VG LLM would translate to much larger base MLLMs (e.g., Llama-3 8B scale)? Is it possible that the spatial reasoning capabilities emerging in larger models would render the explicit geometry encoder less critical?

4. During the training of VG LLM, are the weights of the 3D visual geometry encoder (VGGT) frozen, or are they fine-tuned along with the MLLM backbone? This is an important implementation detail for reproducibility and for understanding the interplay between the two components.

5. The ablation in Table 9 shows that fine-tuning on the mixed dataset already provides a substantial boost in spatial reasoning. Could you further disentangle the contributions of the SPAR-7M dataset and the LaVA-Video-178K dataset? For instance, how does the baseline model (without the geometry encoder) perform when trained only on the LaVA-Video-178K, and how does your VG LLM perform on that same subset? This could help clarify how much of the gain comes from the data.

**Ethical Concerns:**

["NO or VERY MINOR ethics concerns only"]

**Final Justification:**

Thank Authors for addressing my previous concerns. I maintain my original score.

**Limitations:**

yes

**Quality:**

3

**Strengths And Weaknesses:**

Strengths:
1. The paper's primary contribution is both simple and highly effective. Move away from the reliance on explicit, dense 3D inputs (e.g., point clouds, depth maps) and instead leverage the VGGT (CVPR Best Paper) to extract implicit 3D geometry priors from raw video.

2. The performance of VG LLM is outstanding across multiple benchmarks. The state-of-the-art result on VSI-Bench, where the 4B model surpasses much larger proprietary models like Gemini-1.5-Pro, is a headline achievement and provides strong evidence for the method's effectiveness in complex spatial reasoning. The substantial gains in 3D video object detection (Table 3, +19.3 Recall, +8.3 F1) also convincingly demonstrate the model's enhanced ability to handle egocentric-allocentric transformations, a key challenge in this domain.

3. The authors have been thorough in their experimental validation. The evaluation spans a diverse set of tasks, from 3D perception (grounding, captioning) to high-level reasoning. This demonstrates the general applicability and robustness of the proposed framework.


Weaknesses:
1. The success of VG LLM appears to be heavily tied to the performance of the VGGT [43] model, a powerful geometry encoder. The contribution might be perceived as effectively demonstrating the power of VGGT as a feature extractor for MLLMs.

2. The proposed method adds a 1B parameter VGGT encoder to the 3B Qwen2.5-VL backbone. This represents a significant (~33%) increase in model size. The paper mentions the training setup (8x H800 GPUs) but does not provide a detailed analysis of the inference-time trade-offs, which is an important consideration for practical deployment.


3. The experiments are conducted on a 4B scale model. While outperforming larger models is a clear strength, it leaves open the question of how this approach scales. Would the relative gains from the geometry encoder diminish if the base MLLM were much larger and more capable (e.g., a 7B, 34B, 70B model)? An exploration or at least a discussion of this scaling dynamic would strengthen the paper.

---

> ### Author Rebuttal · Authors · 2025-07-31
>
> Thanks for your positive comments!
>
> * **(W1, Q1) Extension to other geometry models.**
>
>     Our framework can also be extended to other geometry models. Specifically, we conducted experiments by replacing the VGGT with Pi3 [1], a powerful visual geometry model with high inference speed.
>
>     The table below gives the comparison results. As depicted in the table, both 'Pi3' and 'VGGT' outperform the baseline ('None') without any geometry encoders. 'Pi3' outperforms 'VGGT' on the 3D visual grounding (ScanRefer), but underperforms it on the 3D dense captioning (Scan2Cap) and 3D video object detection task. This is because ScanRefer requires the model to predict the frame in which an object appears and its associated bounding box, and Pi3's reference-free design makes it more effective for adapting to this task. However, for Scan2Cap and 3D video object detection, models with a fixed reference frame, such as VGGT, performs better.
>
>     |         | ScanRefer |         | Scan2Cap |         | 3D Video Object Detection (4 frames) |         |         |
>     |---------|-----------|---------|----------|---------|-------------------------------------|---------|---------|
>     |         | Acc@0.25  | Acc@0.5 | C@0.5    | B-4@0.5 | Precision@0.25                      | Recall@0.25 | F1@0.25 |
>     | None    | 31.9      | 9.3     | 58.0     | 36.3    | 32.6                                | 27.9    | 29.9    |
>     | Pi3     | 39.7      | 14.7    | 76.4     | 40.0    | 41.3                                | 35.8    | 38.2    |
>     | VGGT    | 36.4      | 11.8    | 78.6     | 40.9    | 41.7                                | 35.7    | 38.2    |
>
> * **(W2, Q2) The inference latency and memory usage of our method.**
>
>     We compared the inference speed of our proposed method with Qwen2.5-VL-3B. The test was conducted on 3D scene understanding tasks, with 100 data samples randomly selected per task, and a maximum of 100 tokens for generation. As presented in the table below, our method incurred an average increase of 20.2% in runtime compared to the baseline. In addition, our method also imposes an additional 36% of memory usage during inference.
>
>     | | Video Object Detection (4 frame) | Video Object Detection (6 frame) | Dense Captioning (16 frame) | Visual grounding (24 frame) |
>     |-------|---|---|---|---|
>     | Qwen2.5-VL-3B | 2.36 | 2.46 | 1.11 | 2.62 |
>     | VG-LLM-4B | 2.73 (+15.6%) | 2.82 (+14.4%) | 1.41 (+27.0%) | 3.24 (+23.6%) |
>
> * **(W3, Q3) Scaling up the LLM backbone.**
>
>     Thanks for your insightful suggestions. We have extended our model to larger language models for spatial reasoning tasks. As shown in the following table, our model built upon Qwen2.5-VL-7B achieved better performance compared to its counterpart without geometry information. This demonstrates that the introduction of geometry information still brings improvements in larger MLLMs.
>
>     | | VSI-Bench | | | | CV-Bench | |
>     |---|---|---|---|---|---|---|
>     | | Abs. Dist | Obj. Size | Room Size | Route Plan | 3D Distance | 3D Depth |
>     | Qwen2.5-VL-3B | 34.7 | 56.4 | 55.0 | 32.0 | 83.5 | 75.8 |
>     | VG-LLM-4B | 37.0 | 56.6 | 60.4 | 34.5 | 88.0 | 84.0 |
>     | Qwen2.5-VL-7B | 36.3 | 55.9 | 58.9 | 28.4 | 87.2 | 87.2 |
>     | VG-LLM-8B | 38.7 | 58.5 | 63.3 | 29.9 | 92.3 | 93.0 |
>
> * **(Q4) Is the 3D geometry encoder frozen or fine-tuned?**
>
>     As detailed in Appendix A.1, we froze the 3D visual geometry encoder during training to preserve its generalization capabilities.
>
> * **(Q5) The ablation study of data composition.**
>
>     We appreciate the reviewer's insightful question regarding our ablation study on data composition. To disentangle the contribution of SPAR and LLaVA-Hound, we trained both the baseline (Qwen2.5-VL-3B) and our model on the SPAR-234K, which utilizes the same data subset as presented in our paper.
>
>     Comparing rows 1 and 2, we can see that the introduction of geometry information brings consistent improvements across most tasks, notably yielding a 4.5% improvement in 3D Distance and a 10.5% improvement in 3D Depth. Furthermore, a comparison between Rows 2 and 4 shows that co-training on a mixture of SPAR-234K and LLaVA-Hound-64K further enhances the model's performance. This is because training exclusively on SPAR-234K can limit the model's generalization ability, whereas the addition of instruction data significantly boosts its robustness.
>
>     | Training Data                 | Model         | Abs. Dist | Obj. Size | Room Size | Route Plan | 3D Distance | 3D Depth |
>     |-------------------------------|---------------|-----------|-----------|-----------|------------|-------------|----------|
>     | SPAR-234K                     | Qwen2.5-VL-3B | 33.5      | 57.3      | 54.0      | 33.0       | 84.8        | 77       |
>     |                               | VG-LLM        | 35.0      | 55.9      | 56.7      | 34.0       | 89.3        | 87.5     |
>     | SPAR-234K + LLaVA-Video-63K   | Qwen2.5-VL-3B | 34.7      | 56.4      | 55.0      | 32.0       | 83.5        | 75.8     |
>     |                               | VG-LLM        | 37.0      | 56.6      | 60.4      | 34.5       | 88.0        | 84.0     |
>
> [1] $\pi^3$: Scalable Permutation-Equivariant Visual Geometry Learning. Wang et al. arXiv 2025.

---

> > ### Comment · Reviewer_MDYF · 2025-08-05
> >
> > Thank you for addressing my previous concerns. I maintain my original score.

---

> > > ### Author Response · Authors · 2025-08-05
> > >
> > > We appreciate your positive feedback and are pleased that our new experiments addressed your concerns. The updated results and clarifications will be included in the revised paper.

---

### Official Review · Reviewer_eRbm · 2025-07-03

**Clarity:** 3
**Significance:** 3
**Originality:** 3
**Rating:** 5
**Confidence:** 4

**Summary:**

The paper aims to integrate 3D priors into the vision-language models. Specifically, the paper proposes to encode the input video images with a feed-forward 3D reconstruction model, VGGT, and enhance the original vlm features with VGGT features via a patch merging strategy. The paper tests this method on several standard 3D benchmarks on visual grounding, captioning and spatial understanding and obtain superior results than other baselines that also do not use sensor 3D information. The ablations show that the proposed strategy of using implicit 3D via features is better than using no 3d information, and also better than directly using the explicit 3D positional coordinates.

**Questions:**

My main concerns are regarding some details and claims like using 3D information via object detectors but claiming not using them as a major point of this work. I would recommend making the numbers of no proposal refinement as the main numbers of table-1, and re-write the description of captioning experiments to be more precise (see my review for additional details). Overall, I think besides these fixable issues, the paper is solid, and I will be happy to support its acceptance.

**Ethical Concerns:**

["NO or VERY MINOR ethics concerns only"]

**Final Justification:**

I am happy to recommend acceptance for this work given the promised clarifications will be added to the camera ready.

**Limitations:**

yes (but authors may consider additional limitations that I mention in my main review)

**Quality:**

3

**Strengths And Weaknesses:**

Strengths:
- The paper is well-written, well-motivated and easy to follow
- The paper shows results on standard benchmarks and uses strong baselines covering all kinds of existing methods — methods which use sensor 3D information, as well as no explicit 3d methods. The ablations are thorough and sufficient.
- The paper tries to benchmark things in this new and important setup where we do not have access to camera pose and depths from sensors.

Weaknesses:
- “Proposal Refinement” term in Table-1 is not defined prior to its introduction. From supplementary, we know that it involves using off-the-shelf 3D object detector and finding the best match detection from it (and refers to lines 224-225 in the main paper). The results with proposal refinement are shows as main result, while the ones without are shown in brackets. I think this is misleading — the main thread in the paper is about not using camera parameters or explicit depth and thus in my opinion the main results should be about that. With proposal refinement, the results indirectly use this information through Mask3D.
    - On a similar note, I would recommend tempering down the use of “ excellent” to describe the performance of the proposed model in Table-1. Even with proposal refinement, the results are much weaker than SOTA, and as I argue above, perhaps the more faithful results are the ones in the brackets which are lower than original ScanRefer’s numbers too. I do not intend to say that not achieving SOTA is a weakness, I appreciate the new setting, but I think it is important to convey the large gap in performance compared to situations when we indeed have access to depth and cameras from our sensors.
- Similar issue as above exists in 3D dense captioning results. Lines 238-239 claim “despite not utilizing 3D camera parameters or explicit depth...” but indeed those numbers utilize them through the 3D detectors that the paper uses. And indeed then the claims that follow become incorrect too like: “indicating the potential of pure-vision solutions for 3d dense captioning” since the proposed method very strongly and clearly relies on 3D detectors. (Side note: “pure vision” solution is a non-standard terminology, many people will argue that using depth and camera is pure vision as well).
    - (Minor comment): L239 saying that the proposed method surpasses prior SOTA LEO is somewhat weird since the same table contains several other methods which are better than both LEO as well as the proposed method. Perhaps consider deleting this statement.
- It’s important to note that methods like VGGT are in fact trained on scannet. this paper studies the use of these models on benchmarks based on 3D scenes that are very much in-distribution of VGGT. Thus showing that 3D tokens help in these benchmarks compared to 2D only methods is riskier — the true test will be in videos / datasets outside VGGT’s distribution. Note: I am not expecting authors to perform such an experiment, but perhaps it would be nice to make this explicit in the limitations.

---

> ### Author Rebuttal · Authors · 2025-07-31
>
> We are really grateful for your constructive suggestions.
>
> * **(W1, Q1) The table presentation of Table 1.**
>
>     We included proposal refinement in our experiments to ensure a fair comparison with prior works like LEO [1] and Chat-Scene [2], which often leverage off-the-shelf 3D object detectors to extract proposals, either directly or indirectly. We agree with you that the table should focus on the results without using any depth or camera information, and we will revise the table presentation in the next version.
>
> * **(W1, W2, Q1) The claims about the 3D visual grounding and 3D dense captioning.**
>     Our framework could also be extended to perform 3D dense captioning by locating object bounding boxes and generating the associated object captions in a single forward pass, akin to 3D video object detection. However, this promising direction is beyond the scope of our current work and will be reserved as future research.
>     Thank you again for helping us improve the clarity and rigor of the paper. We will revise the description according to your suggestions in the next version.
>
> * **(W3) Benchmarks outside VGGT's distribution.**
>     Given that VGGT has already leveraged almost all existing 3D scene datasets, there is a scarcity of available OOD evaluation benchmarks. Therefore, we will explore the generalization of our method as future work and discuss this limitation in the revised version.
>
> [1] An Embodied Generalist Agent in 3D World. Huang et al. ICML 2024.
>
> [2] Chat-Scene: Bridging 3D Scene and Large Language Models with Object Identifiers. Huang et al. NeurIPS 2024.

---

> > ### Comment · Reviewer_eRbm · 2025-08-03
> >
> > Thank you to authors for their response. I am happy to recommend acceptance for this work given the promised clarifications will be added to the camera ready.

---

> > > ### Author Response · Authors · 2025-08-04
> > >
> > > Thanks for your positive feedback and recommendation. In the revised manuscript, we will update all the comments into the final version.

---

### Official Review · Reviewer_dMC1 · 2025-07-03

**Clarity:** 3
**Significance:** 2
**Originality:** 2
**Rating:** 4
**Confidence:** 4

**Summary:**

- The paper proposes Video-3D Geometry LLM (VG LLM), a framework that explicitly integrates 3D visual geometry priors into Multimodal Large Language Models (MLLMs) for enhanced 3D spatial understanding and reasoning.
- The paper conducted extensive experiments showing the effectiveness of incorporating the features from a pretrained 3D visual geometry encoder.

**Questions:**

- I wonder if the authors have tried plugging predicted 3D information (from off-the-shelf models) into baseline models that require 3D inputs? This would create a fair and controlled comparison between VG LLM and existing 3D-enhanced baselines

**Ethical Concerns:**

["NO or VERY MINOR ethics concerns only"]

**Final Justification:**

After reading the other reviews and the authors' response, I still believe that the novelty of the proposed method is relatively limited and cannot be fully offset by the empirical results. However, I appreciate the authors' efforts in the rebuttal, as they have addressed most of my concerns. Therefore, I am willing to increase my score to 4.

**Limitations:**

yes

**Quality:**

3

**Strengths And Weaknesses:**

**Strengths**
- The paper proposes to incorporate the 3D features from a pretrained 3D visual geometry encoder into MLLMs for better 3D spatial understanding and reasoning.
- The proposed model demonstrates notable improvements on a range of 3D scene understanding and spatial reasoning benchmarks without requiring explicit 3D input.

**Weaknesses**
- From an architectural standpoint,  it seems like the novelty is mainly in leveraging 3D representations from a pretrained 3D visual geometry encoder (developed in prior work) and integrating these with 2D visual tokens, which is relatively limited.
- The paper lacks analysis or investigation of alternative fusion strategies for integrating 3D geometry and 2D visual tokens.
- It is unclear to me how the experiments in Table 8 can be used to show that “the improvement in spatial reasoning and 3D scene understanding in MLLMs 308 is genuinely driven by visual geometry modeling”.  Given that the camera info, etc. is predicted from visual geometry features and this prediction is inherently imperfect, it is natural for the pred. XX variants underperform the original visual geometry features.

---

> ### Author Rebuttal · Authors · 2025-07-31
>
> Thanks for your time, effort, and valuable comments!
>
> * **(W1) The novelty of our method.**
>
>     We highlight our core contributions as follows:
>     1. We propose a novel framework to enhance spatial understanding in MLLMs, incorporating a 3D geometry encoder that provides latent geometry information from video inputs. Unlike previous approaches heavily dependent on large volumes of 3D data, our method efficiently processes only video sequences, thereby creating new opportunities for significantly boosting the 3D understanding and spatial reasoning capabilities of MLLMs.
>     2. While straightforward, our extensive experimental results unveil the potential of leveraging geometry models for improved 3D understanding and spatial reasoning. We pioneered a new setting for 3D scene understanding by eliminating the dependence of 3D point clouds in MLLMs and validated the superiority of our methods across tasks, with a substantial improvement of F1 of 8.3 on 3D video object detection. Moreover, our method attains an impressive average score of 46.1 on VSI-Bench, a benchmark demanding complex spatial reasoning, outperforming previous state-of-the-art methods.
>
>
> * **(W2) The effect of feature fusion strategies.**
>
>     We have experimented with several feature fusion strategies on 3D scene understanding tasks.
>
>     1. 'cross-attention': We refer to the strategy in PAVE [1], which consists of multiple blocks of cross-attention modules and MLPs with skip connections. 2D visual tokens serve as queries, while 3D visual tokens serve as keys and values. 2D positional embeddings are added to both the queries and keys.
>     2. 'concat': This approach first concatenates the 2D and 3D visual tokens along the feature dimension, followed by an MLP to transform them into the text embedding space.
>     3. 'add': This is the strategy employed in our paper. It directly adds the 2D and 3D visual tokens at a patch level.
>
>     As the table below illustrates, the 'cross-attention' strategy significantly improved performance over the baseline. Moreover, increasing the number of layers to three yielded even greater performance gains. While 'concat' performed worse than the baseline on ScanRefer, it outperformed it on Scan2Cap and 3D video object detection. Nevertheless, 'add' surpassed all other comparison methods despite its simplicity.
>
>     | | ScanRefer | | Scan2Cap | | 3D Video Object Detection (4frames) | | |
>     |---|---|---|---|---|---|---|---|
>     | | IoU@0.25 | IoU@0.5 | C@0.5 | B-4@0.5 | Precision@0.25 | Recall@0.25 | F1@0.25 |
>     | Baseline (w/o geometry) | 31.9 | 9.3 | 58.0 | 36.3 | 32.6 | 27.9 | 29.9 |
>     | Concat | 27.7 | 6.8 | 75.7 | 40.0 | 37.1 | 32.5 | 34.4 |
>     | Cross-attention (1 layer) | 33.7 | 10.7 | 74.9 | 40.1 | 38.0 | 33.6 | 35.4 |
>     | Cross-attention (3 layers) | 34.4 | 10.5 | 75.5 | 40.2 | 38.5 | 33.0 | 35.4 |
>     | Add | 36.4 | 11.8 | 78.6 | 40.9 | 41.7 | 35.7 | 38.2 |
>
> * **(W3) The clarification of Table 8.**
>
>     The table primarily compares model variants that incorporate different types of spatial signals.
>     1. The most crucial comparison is between our model (last row) and the baseline without any geometry information (first row). The significant improvement demonstrates the effectiveness of incorporating 3D geometry signals into MLLMs.
>     2. We agree with the reviewer that the predicted attributes (pred. XX) are inherently imperfect. The core question behind these variants was to determine the most effective way to integrate spatial signals into the MLLM: explicit, strucutred attributes (e.g., predicted camera pose, pointmaps) or implicit, dense features? This question is non-trivial. The pred. XX attributes, while structured, suffer from prediction errors. Conversely, the visual geometry features, while dense, are learned without direct supervision. Furthermore, the predicted attributes are not solely predicted from the geometry features but generated by a dense prediction transformer [2] that fuses multi-level hidden states with multi-scale DINO image features. Therefore, it was not initially obvious which form of spatial signal would be more beneficial. Our empirical results conclude that the visual geometry features make a more direct and effective contribution to the model's 3D understanding capabilities.
>
>     These two points confirm the importance of our 3D geometry modeling.
>
> * **(Q1) Baselines that incorporate 3D information from off-the-shelf models.**
>
>     Video-3D LLM [3] introduces a method for 3D understanding in MLLMs by directly injecting pointmaps into visual representations. Following this approach, and as detailed in Table 8, we integrated pointmap predictions from VGGT—the current state-of-the-art model for tasks like pointmap prediction and camera pose estimation—into our LLM, which we refer to as 'Pred Point Info.' Our experimental results clearly indicate that incorporating explicit pointmap information improves performance in tasks such as dense captioning and 3D video object detection. Nevertheless, our method significantly outperforms these comparison methods with explicit geometry information.
>
> [1] PAVE: Patching and Adapting Video Large Language Models. Liu et al. CVPR 2025.
>
> [2] Vision Transformers for Dense Prediction. Ranftl et al. ICCV 2021.
>
> [3] Video-3D LLM: Learning Position-Aware Video Representation for 3D Scene Understanding. Zheng et al. CVPR 2025.

---

### Comment · Area_Chair_YB8X · 2025-08-05
**Reminder: Please follow up on the authors’ rebuttal**

Dear Reviewers,

Many thanks to Reviewers eRbm and MDYF for the thoughtful discussion so far.

Reviewers dMC1and FeDL, just a gentle reminder – we’d greatly appreciate it if you could take a moment to review the authors’ responses and share any follow-up thoughts.

With the discussion phase wrapping up soon, your timely input can help ensure a productive exchange and give the authors a final chance to clarify any remaining points.
Thanks again for your time and contributions to the NeurIPS review process.

Best,

Your AC

---

### Decision · Program_Chairs · 2025-09-17

**Decision:**

Accept (poster)

**Comment:**

All reviewers agreed that the problem under investigation is significant, the proposed approach is effective, and the results are encouraging. The rebuttal adequately addressed most of the reviewers’ concerns, leading all of them to recommend acceptance. The authors are advised to further refine the final version of the paper by incorporating the reviewers’ feedback.